# The Strategy of the Brain to Maintain the Force Production in Painful Contractions—A Motor Units Pool Reorganization

**DOI:** 10.3390/cells11203299

**Published:** 2022-10-20

**Authors:** Klaus Becker, Márcio Goethel, Pedro Fonseca, João Paulo Vilas-Boas, Ulysses Ervilha

**Affiliations:** 1Porto Biomechanics Laboratory, University of Porto, 4200-450 Porto, Portugal; 2Center of Research, Education, Innovation and Intervention in Sport, Faculty of Sport, University of Porto, 4200-450 Porto, Portugal; 3Laboratory of Physical Activity Sciences, School of Arts, Sciences, and Humanities, University of São Paulo, São Paulo 03828-000, Brazil

**Keywords:** experimental pain, hypertonic saline, motor unit, muscle, pain models

## Abstract

A common symptom in neuromuscular diseases is pain, which changes human movement in many ways. Using the decomposed electromyographic signal, we investigate the strategy of the brain in recruiting different pools of motor units (MUs) to produce torque during induced muscle pain in terms of firing rate (FR), recruitment threshold (RT) and action potential amplitude (MUAP_AMP_). These properties were used to define two groups (G1/G2) based on a K-means clusterization method. A 2.0 mL intramuscular hypertonic (6%) or isotonic (0.9%) saline solution was injected to induce pain or act as a placebo during isometric and isokinetic knee extension contractions. While isometric torque decreases after pain induction with hypertonic solution, this does not occur in isokinetic torque. This occurs because the MUs re-organized after the injection of both solutions. This is supported by an increase in RT, in both G1 and G2 MUs. However, when inducing pain with the hypertonic solution, RT increase is exacerbated. In this condition, FR also decreases, while MUAP_AMP_ increases only for G1 MUs. Therefore, this study proposes that the strategy for maintaining force production during pain is to recruit MUs with higher RT and MUAP_AMP_.

## 1. Introduction

Neuromuscular diseases affect the peripheral nervous system, which includes motor and sensory neurons, the muscle itself and the neuromuscular junction. While some neuromuscular diseases are also associated with the central nervous system, most are restricted to the peripheral nervous system [1]. Moreover, neuromuscular diseases comprise a wide set of syndromes, in which muscle pain is a symptom frequently present and potentially associated with movement impairment [2]. As a multidimensional symptom, pain study may prove beneficial to reduce clinical bias and improve its understanding. Experimental muscle pain can be induced by intramuscular injections of hypertonic saline solution [3,4] and has been used to study its effect on motor control. It is known that pain can cause multiple levels of change in muscle activation; however, there is no definitive explanation for how pain alters the motor unit (MU) recruitment strategy [5]. It has been shown that muscle pain interferes with force production by changing the motoneuron discharge in a central level [6]. This may be caused by the fact that noxious stimuli may interfere with excitatory or inhibitory inputs to the motoneuron drive [7]. However, some studies demonstrated that pain does not interfere with the ability to perform a force production task [8,9], which indicates that the motor system must have mechanisms to compensate for the changes caused by pain in the muscle’s electrical activity [10,11].

The muscle’s electrical activity is studied by means of surface electromyography (EMG) and it has the potential to provide relevant information regarding pain’s peripheral and central myoelectric manifestations and ensuing neuromuscular impairments [12,13]. Although EMG amplitude and force production during maximal voluntary contractions (MVCs) are decreased during muscle pain conditions [14], the scientific literature still presents conflicting results that support theories of pain repercussions on muscle activation [5]. Some studies support the idea of an increasing muscle activation due to pain, leading to a vicious cycle of continued pain [15,16], while others display a muscle inhibition, with increasing activation of the antagonist muscle [17,18]. This entails the possibility that reduced muscle activation during pain may be accompanied by other changes in motor control in order to maintain continuous force production.

Currently, the scientific literature presents two hypotheses to explain the motor unit activation strategies in response to pain, while sustaining a constant force production. The first hypothesized that, at high forces, the central nervous system will increase the synaptic input received by high-recruitment-threshold (RT) motor units, to compensate for the inhibition of low-RT motor units [11]. The second hypothesis is that pain implies an activity redistribution between motor units, with a preference for larger motor units. A possible benefit of this mechanism is the unloading fibers perceived as painfully signalized [19,20]. Although the MU redistribution might not explain the behavior of every muscle [21], a shift in the MU pool recruitment pattern and RT of specific MUs [22,23] may be a plausible mechanism for force maintenance. 

The purpose of this study was to investigate the changes in the pool of MU activation, such as RT, FR and motor unit action potential amplitude (MUAP_AMP_) during pain. To investigate the effect of acute muscle pain on the motor drive, we used EMG decomposition (dEMG), which allows for access to the MU firing rate (FR), RT and MUAP_AMP_. To ensure a broad spectrum of MU recruitment, we decided to investigate the adaptations in the MU pool during maximum isokinetic contractions.

## 2. Materials and Methods

### 2.1. Participants

Fourteen male participants (28 ± 5 years, 175.8 ± 4.7 cm, 74.4 ± 11 kg, 24.0 ± 2.9 body mass index) were recruited. The inclusion criteria to participate in this study were to be free from any musculoskeletal impairment in the last 3 months, to have experienced no pain in the last 7 days or no continuous pain for more than 90 days in the last 3 years. The study was conducted in accordance with the Helsinki Declaration, with all participants having read and signed informed consent prior to data collection. This study was approved by the Ethics Committee of the local institution with the protocol number 25/2019.

### 2.2. Induced Muscle Pain

Each participant visited the laboratory twice, with seven days between visits. A randomized cross-over design was utilized, with participants receiving either a single bolus of 2 mL hypertonic (6.0%) saline solution, designed to induce pain, or an isotonic (0.9%) saline solution, designed to act as a placebo. The solutions were injected into vastus lateralis muscle belly in each visit. A 5-milliliter disposable syringe and disposable 40 × 0.8 mm stainless-steel needle (BD Emerald, Fraga España, Spain) were used. Baseline (before injection) and immediately after injection, responses to intramuscular (hypertonic or isotonic) injection were evaluated. Pain intensity was reported by the participant every minute from the injection by using a visual analog scale (VAS) anchored with “no pain” at zero and “worst pain imaginable” at 10.

### 2.3. Muscular Assessment

The volunteers were comfortably seated on the chair of an isokinetic dynamometer (Biodex Multi-Joint System 4, Biodex Medical System, Shirley, NY, USA) and the belt and thoracic straps were used to keep the volunteer immobilized according to the manufacturer guidelines. In each session the volunteers performed: (i) a specific warm-up of 10 concentric/eccentric isokinetic contractions of knee extension at 60°/s, with a range of motion of 60°, from 90° to 30°, (0° = full extension); (ii) two repetitions, 60 s apart, of 3 s maximal voluntary isometric contractions of knee extension, with the knee joint at an angle of 60º (0° = full extension); (iii) 5 maximum-effort knee extensions (concentric/eccentric) at the same velocity and range of motion as step 1, with strong verbal encouraging from the researcher; (iv) a 3 min rest period; (v) the injection of the hypertonic or isotonic solution; and (vi) the repetition of steps (ii) and (iii) immediately after saline injection. The selection of the type of saline injection in the first session was random. All tests were performed using the dominant lower limb, defined by the leg used to kick a ball.

### 2.4. EMG Recording and Decomposition

The signal was acquired using the Delsys Trigno^TM^ system with a dEMG Galileo sensor (Delsys, Natick, MA, USA), as previously described by Priego-Quesada et al. [24]. The recommendations of the SENIAM project were followed regarding muscle surface preparation and the sensor placed on the belly of the *vastus lateralis* muscle, in the same direction of the muscle fibers. Data collection was performed using the Delsys EMGWorks 4.8.0 software (Delsys, Natick, MA, USA).

The decomposition of the electromyographic signal was performed through the algorithm created by De Luca [25] and improved by Nawab [26], by means of the NeuroMap System software (Delsys, Natick, MA, USA). For each detected motor unit, its RT (MU onset, presented as the percentage of the torque of MVIC), FR and MUAP_AMP_ were considered for analysis. The minimum accuracy of 80% was defined for MU detection.

### 2.5. Clusterization Method

Using the K-means approach in MATLAB software (R2022a, Natick, MA, USA) was possible to group the MUs [27] into clusters based on three variables: RT, FR and MUAP_AMP_. In a gross analogy of the fiber type, we pre-defined the number of clusters for two: G1 and G2.

### 2.6. Statistical Analysis

The normality of data distribution was verified with the Shapiro–Wilk test. Results are presented by mean and standard deviation. A repeated-measures MANOVA was applied to test the effects between epochs (pre and post), conditions (hypertonic and isotonic) and MU groups G1 and G2. When main effects were found, multiple comparisons using Bonferroni correction were conducted. Then, each MVIC and peak torque in concentric and eccentric phases was compared using the Paired Student’s *t*-test. Cohen’s *d* criteria were used to calculate the power of analyses (>0.2: small; >0.50: moderate; >0.80: large) [28]. All tests were performed using SPSS Statistics version 27 (IBM Corporation, Armonk, NY, USA) and significance was set at *p* < 0.05. The results’ statistical power was assessed a posteriori using G*Power v3.1 (RRID:SCR_013726).

## 3. Results

### 3.1. Experimental Muscle Pain

The average VAS pain score in response to the injection of hypertonic and isotonic saline solution is illustrated in Figure 1. The painful sensation was different between the hypertonic and isotonic injections. With the hypertonic solution, the painful sensation lasted for the full set of contractions, reaching its peak 2 min after the injection and ceasing completely after 7 min. All participants reported that the pain was only felt in the injection site.

### 3.2. The Effect of Pain in Torque Production

The torque generated at the MVIC decreased after pain induction, but did not change after the placebo injection. Figure 2 presents these results as the torque loss (%MVIC) after injection. Regarding the isokinetic torque, no difference was found during each epoch.

### 3.3. Motor Unit Decomposition

All MUs were detected during the five isokinetic contractions. A total of 365 and 481 motor units could be tracked across isotonic and hypertonic conditions, respectively, from all participants during pre-injection and post-injection epochs. As depicted in Table 1, the number of MUs recruited in G1 during pain expressively increased, relative to its pre-injection epoch; this phenomenon cannot be seen in other groups or epochs.

### 3.4. Clusterization of Motor Units

The RT was the most important variable for the clusterization of motor units, as shown in Figure 3. Since the method for clusterization of MU used three variables, a three-dimensional plot would be ideal to illustrate their iteration. However, this is impracticable for this means of communication, so we decided to produce several two-dimensional plots containing different combinations of variables to elucidate the separation effect caused by RT. It is interesting to notice how the absence of RT during cluster analysis produces a mixed effect on the MU, independently of injection and moment.

### 3.5. Motor Unit Variables

In G1, the two-way MANOVA test showed a significant FR decreased between epochs with the hypertonic injection (pre: 7.21 ± 3.81 pps vs. post: 6.01 ± 3.46 pps. *p* < 0.016, power (1-β) = 0.7750213), while no difference was observed with the isotonic solution (pre: 7.47 ± 3.4 pps vs. post: 7.07 ± 2.8 pps. *p* < 0.409, power (1-β) = 0.0709543) (Figure 4a). The RT decreased between epochs with hypertonic injection (pre: 11.55 ± 6.39% vs. post: 19.73 ± 10.25%. *p* < 0.001, power (1-β) = 0.9910023) and no difference occurred during placebo injection (pre: 13.01 ± 6.8 vs. post: 15.74% ± 8.9. *p* = 0.06, power (1-β) = 0.9228522) (Figure 4b). The MUAP_AMP_ increased between epochs during hypertonic injection (pre: 0.16 ± 0.09 mV vs. post: 0.22 ± 0.48 mV, *p* < 0.001, power (1-β) = 0.916157) and there were no differences between epochs during placebo injection (pre: 0.21 ± 0.14 mV vs. post: 0.20 ± 0.17 mV, *p* = 0.32, power (1-β) = 0.9228522). Differences found in this variable are due to clustering motor units of higher amplitudes that decreased their RT (see Figure 4).

In G2, the FR decreased during hypertonic (pre: 4.79 ± 3.01 pps vs. post: 3 ± 2.12 pps. *p* < 0.001, power (1-β) = 0.958708) and placebo injections (pre: 5.79 ± 3.26 pps vs. post: 4.93 ± 2.69 pps. *p* = 0.008, power (1-β) = 0.3410492) (Figure 4a). The RT of G2 increased during hypertonic (pre: 43.34 ± 12.89% vs. post: 58.23 ± 17.51%. *p* < 0.001, power (1-β) = 0.9969426) and isotonic injection (pre: 39.20 ± 10.93% vs. post: 46.63 ± 12.71%. *p* < 0.001, power (1-β) = 0.6590949) (Figure 4b). The MUAP_AMP_ did not change during either injection. Figure 4c shows a large increase in amplitude of G1 MUs.

## 4. Discussion

Changes in the muscle activation caused by acute pain seems to be task dependent and resulting in decreased muscle activation [18,29]. However, there are studies contradicting this finding, showing both the agonist and antagonist decreased myoelectric activity while the subjects are still able to complete a force-demanding task [9,14]. In this study, results help to understand the adaptations that pain imposes to the motor control system. Moreover, it is in line with studies showing that experimentally induced muscle pain in humans decreases maximal isometric force [17,30], though, interestingly, it does not change maximal isokinetic force.

In the current study, we investigated the recruitment of MUs on vastus lateralis muscle during maximum isokinetic contractions before and during induced acute muscle pain. Our hypothesis was that during pain, there is a decreased recruitment demand for the preferential muscle fibers in order to prevent further damage on those fibers. This is possible by recruiting motor units of increased thickness that have higher RT and MUAP_AMP_ as well as lower FR. Among these three variables, RT is the one that indicates whether a particular motor unit will be recruited or not. Therefore, changes in the RT due to pain indicates that the motor system adopted a different strategy of MU recruitment. This was observed in our study, during the isokinetic task, while the maximum force performed during the execution of the task was not altered.

The data show that RT and FR are variables significantly altered by pain, while RT presented an increase, FR presented a decrease due to pain, despite the force produced remaining unaffected. Increases in RT plus maintenance of the produced force suggest the reorganization of the motor system, with consequent greater recruitment of larger MUs [31]. This is the mechanism that is supported by two studies, where evidence was presented that acute pain evokes greater recruitment of motor units with a higher recruitment threshold, whether the motor task is of high [11] or low [19] demand of force. It has also been suggested in the literature that the pain can induce the motor system to recruit previous quiescent MUs [8,10], as well as that the voluntary excitability of each MU might change accordingly with its RT [11]. Our data support the idea that the motor system recruits new MUs, as shown by the increased RT, due to acute pain, in both groups. MUs with higher RT generally also have lower FR, which explains the overall decrease in FR during pain. Our findings also suggest that those MUs of higher RT and higher MUAP_AMP_ were recruited earlier in order to maintain the isokinetic force production, since those MUs have lower mean FR. This reasoning explains why MUAP_AMP_ increased only in the G1 group, since this is the group with lower RT mean and generally MU of higher MUAP_AMP_ have high RT.

Our data are supported by two recent studies. Hodges et al. (2021) investigated the time-course of motoneurons after hyperpolarization on a single motor unit activity. Their data showed shortened and lengthened afterhyperpolarization time-course [19], which indicates a non-uniform effect of nociception on motoneuron excitability. In another study, Cleary et al. (2022) showed that acute experimental pain alters the quadricep motor unit discharge rate when pain is induced in tibialis anterior, which does not contribute mechanically with quadricep force production or biceps femoris (antagonist muscle). The discharge rate decreased in vastus lateralis and vastus medialis during pain, independently of the painful muscle while the force remained the same, which indicates a redistribution of motor unit activity, with some units recruited in control or pain but not both [20].

In conclusion, our study provides evidence that the motor system performs a selective recruitment of MUs during muscle pain in order to maintain the same isokinetic torque production. To do so, MUs with higher RT and MUAP_AMP_ are recruited, even if the FR decreases. This does not support the hypothesis of a redistribution of activity in a single MU, explaining the entire response to painful stimuli.

## Figures and Tables

**Figure 1 cells-11-03299-f001:**
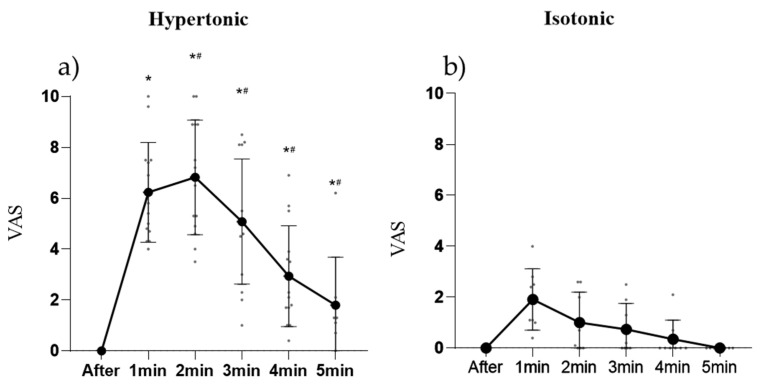
Pain score obtained from a visual analog scale, where “0” means no pain and 10 “maximal tolerable pain”. (**a**) Pain curve during hypertonic saline injection (pain induction). (**b**) Pain curve during isotonic saline injection (placebo). * Significant difference from baseline (*p* < 0.05), # Significant difference from the previous minute (*p* < 0.05).

**Figure 2 cells-11-03299-f002:**
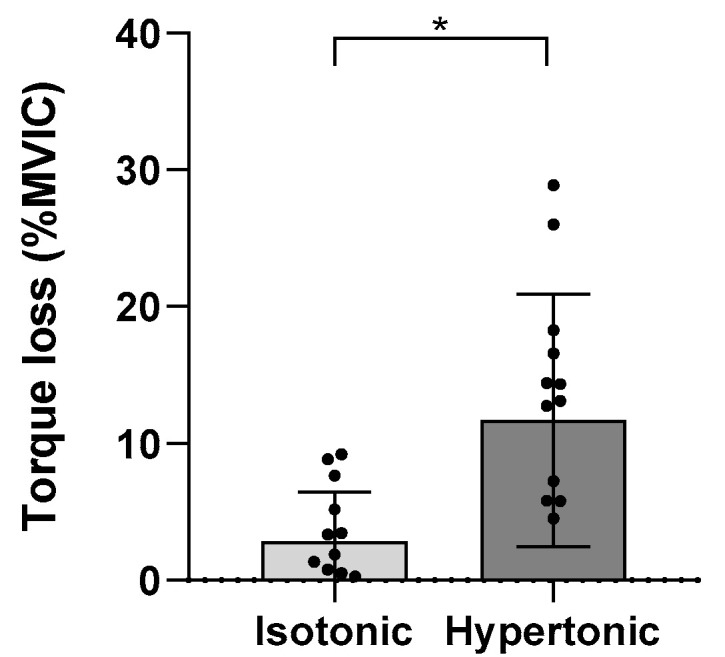
Torque loss after injection with isotonic and hypertonic solutions when compared to torque produced during pre-injection MVIC. * Significant difference between saline solutions (*p* < 0.05).

**Figure 3 cells-11-03299-f003:**
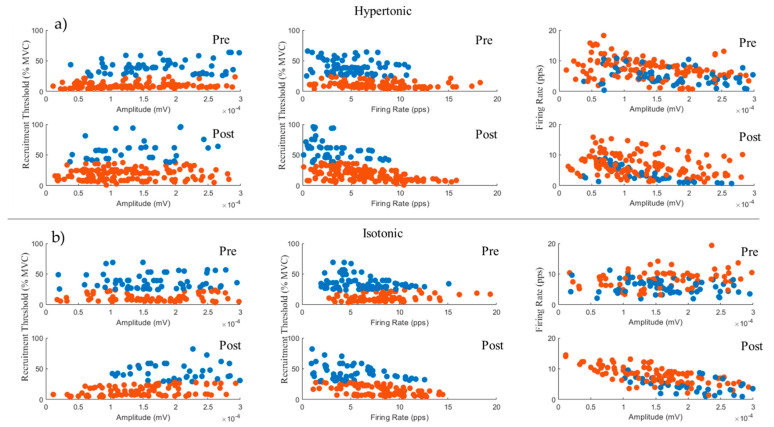
Two-dimensional plots of the 3 variables considered in the clustering analyses. (**a**) Clusters of MUs during induced muscle pain in each epoch. (**b**) Clusters of MUs during placebo protocol in each epoch. Each point represents a motor unit, identified as belonging to group 1 (red) or 2 (blue), according to the recruitment threshold value. Pre- and post-injection values are shown.

**Figure 4 cells-11-03299-f004:**
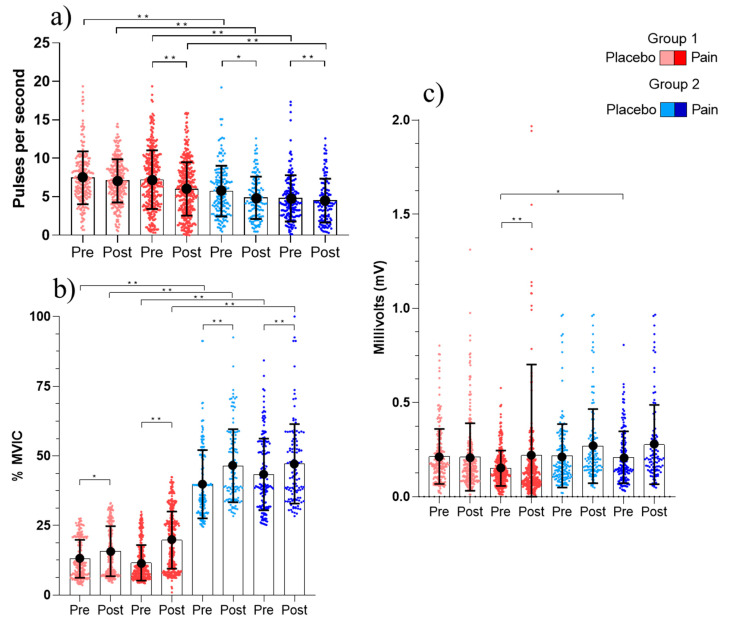
Mean and standard deviation of motor unit variables identified in nineteen subjects during five isokinetic contractions and stratified in epoch, solution type and MU groups. (**a**) Mean firing rate (**b**) recruitment threshold (**c**) MUAP_AMP_. Data were obtained during baseline (pre) and during (post) experimentally induced pain or placebo. Statistical significance was tested using MANOVA between moments in the same condition (* *p* < 0.01, ** *p* < 0.001).

**Table 1 cells-11-03299-t001:** Number of motor units (MUs) recognized during five isokinetic contractions, separated by groups and epochs in each injection protocol. The table shows the total motor units of the sample from each epoch above the accuracy cutoff of 80%.

	G1		G2		Total MUs
	Pre	Post	Pre	Post	
**Hypertonic**	148	180	89	64	481
**Isotonic**	98	115	87	65	365

## Data Availability

To request data, contact us by email: klausmagnobecker@gmail.com.

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
