# Peer review of "The Strategy of the Brain to Maintain the Force Production in Painful Contractions—A Motor Units Pool Reorganization"

_cells, 2022, doi:10.3390/cells11203299_

Round 1
Reviewer 1 Report
In this report, Becker et al investigated how the properties of motor units are modified by muscular pain, Although mainly descriptive, the power of the experimental design and the analysis together with the writing style (concise and clear)make the report of an excellent piece of work. It provides new insights into brain adaptations in a very relevant problem from a clinical perspective.
My major point is about the sample used in this study. The overall number of participants is relatively low (19) and there is a strong sex bias (14 males and 5 females). Finally, it seems (from the mean height and weight) that the cohort also encompasses participants with a high BMI
If possible in terms of study enrolment, including more participants in the(5-10 additional females) would be a must and would considerably strengthen the results/conclusions of this study. In addition, it would be beneficial to provide a table detailing the age, sex, height and weight for each participant.
Author Response
In this report, Becker et al investigated how the properties of motor units are modified by muscular pain, Although mainly descriptive, the power of the experimental design and the analysis together with the writing style (concise and clear) make the report of an excellent piece of work. It provides new insights into brain adaptations in a very relevant problem from a clinical perspective.
We thank the reviewer for this assessment.
My major point is about the sample used in this study. The overall number of participants is relatively low (19) and there is a strong sex bias (14 males and 5 females). Finally, it seems (from the mean height and weight) that the cohort also encompasses participants with a high BMI
As suggested, we added information regarding the sample’s average BMI, height and weight information in the first paragraph of the Materials and Methods section. Although we recognize the sample include overweight individuals (BMI > 24.9) it also has normal BMI, as denoted by the BMI average value of 24.
If possible, in terms of study enrolment, including more participants in the (5-10 additional females) would be a must and would considerably strengthen the results/conclusions of this study. In addition, it would be beneficial to provide a table detailing the age, sex, height and weight for each participant.
We thank the reviewer for pointing the potential issue with the gender difference of our sample. Not being possible, at this moment, to add female participants to this manuscript, we decided to exclude the female group and present only the results for the 14 male participants. This decision was also supported by a paper that recently (after submission) came to our attention stating that male and female individuals may have significant differences in terms of MU firing rate (published October 2022, doi: 10.1016/j.jelekin.2022.102689).
We recognize that a sample reduction is not desirable and may have an influence in the statistical results and their power. Therefore, and in order to guarantee that our statistical results are reliable, we performed a statistical power analysis, which can now be found presented along with the other statistical results.
Considering the suggestion of adding a table with the sample’s individual values of mass, height and age, we think that since our approach was to pool the motor units of all the participants of this study, and analyze their behavior (as a group). We don’t think that detailing the individual characteristics of each participant would improve our data reporting, or present relevant information, when mean and standard deviation information (of the group) is already provided.
Reviewer 2 Report
The authors wrote about the axis of muscle and brain mechanisms of pain. He processes the problem well. The number of elements is low, but this does not detract from the value of the thesis. The hypothesis is nicely developed and shaped.
I miss the MRI examinations from the pain examinations.
The figures are easy to interpret, but I would recommend presenting the results in a table instead of figures 2 and 3.
I congratulate you on your thesis.
Author Response
The authors wrote about the axis of muscle and brain mechanisms of pain. He processes the problem well. The number of elements is low, but this does not detract from the value of the thesis. The hypothesis is nicely developed and shaped.
We thank the reviewer for this assessment.
I miss the MRI examinations from the pain examinations.
The induced muscle pain protocol has a short window of opportunity to perform measurements. The use of an MRI machine would have increased the complexity of the experimental setup, as it is not possible to have the participant simultaneously at the isokinetic dynamometer and the MRI machine. Additionally, and unfortunately, an MRI machine is something that at the time we don’t have access to.
The figures are easy to interpret, but I would recommend presenting the results in a table instead of figures 2 and 3.
As suggested, we have converted figure 3 into a table (Table 1) but, according with other reviewer suggestion, we decided to keep the information of figure 2 as a bar scatter to denote the data points range.
I congratulate you on your thesis.
We thank the reviewer.
Reviewer 3 Report
The authors assessed firing rate, thresholds and motor unit amplitudes in human volunteers in a hypertonic saline induced muscle pain model to find out how/why muscle strengths is maintained during pain. They used EMG studies to assess physiologic parameters and pain ratings on a visual analog scale to quantify the effect of hypertonic saline. The study is clinically relevant considering the high prevalence of e.g. low back pain and is novel and the text is mostly clear. However, the presentation of the data is disappointing and key messages e.g. in the abstract are lost / unclear in part owing to the use of multiple abbreviations. The conclusion (end of discussion) is unclear for me. The figures 1 and 2 show very high variability, and individual data points are not revealed (scatter plots or bar scatter or box/ scatter or multiple line graphs (fig1) should be used to reveal sample sizes and biological variability. In Figure 3 one does not know if the figure shows the total numbers of all subjects. Figure 5 is also bar charts not showing individual data points and should be replaced with scatter plots Figure 3: hypertonic saline and isotonic saline look fairly alike. hollow and black dots need further explanation. What was the threshold to set it as black? How justified? One would like to see an association analysis of individual VAS versus firing rate, threshold and amplitude. Were there differences between male and female subjects?Author Response
The authors assessed firing rate, thresholds and motor unit amplitudes in human volunteers in a hypertonic saline induced muscle pain model to find out how/why muscle strengths is maintained during pain. They used EMG studies to assess physiologic parameters and pain ratings on a visual analog scale to quantify the effect of hypertonic saline. The study is clinically relevant considering the high prevalence of e.g. low back pain and is novel and the text is mostly clear.
We thank the reviewer for this assessment.
However, the presentation of the data is disappointing and key messages e.g. in the abstract are lost / unclear in part owing to the use of multiple abbreviations.
Since we used multiple techniques as induced muscle pain, dynamometry, electromyography decomposition and cluster analysis the abstract became too short for a clearer communication. Our effort in the abstract was to communicate most important findings along with a sentence with the main hypothesis. However, we have made an effort to improve our abstract by removing the amount of information regarding our methodological approach, and expanding the results. Unfortunately, it was not possible to reduce the number of abbreviations due to their importance to the information being communicated. However, each abbreviation meaning is introduced in the abstract.
The conclusion (end of discussion) is unclear for me.
We appreciate the reviewer for the critics, we think we clarify the text to a more objective and succinct conclusion. In the scientific literature there is currently two theories of motor units adaptations in response to pain, one indicates that there is a non-uniform inhibition in the whole pool of motor units, while the other suggests that the firing rate of low threshold motor units decreases while and increases in high threshold motor units. Our data supports the non-uniform inhibition and indicates that to underpin the same force production the motor system select a different pool of motor units.
We alter the conclusion in order to be more clarified
The figures 1 and 2 show very high variability, and individual data points are not revealed (scatter plots or bar scatter or box/ scatter or multiple line graphs (fig1) should be used to reveal sample sizes and biological variability.
We thank the reviewer’s suggestion. We added individual scatter data to the previous bar graphs of figure 1 and 2.
In Figure 3 one does not know if the figure shows the total numbers of all subjects.
Following other reviewer suggestion, we converted Figure 2 into a table (Table 1) and clarified this matter in the legend.
Figure 5 is also bar charts not showing individual data points and should be replaced with scatter plots
We thank the reviewer for this suggestion. In order to have a compromise between the simplicity of a bar graph, and the dispersion information of a scatter plot, we converted Figure 5 to a scatter bar graph. We also added colors to each group and condition to make them easier to differentiate.
Figure 3: hypertonic saline and isotonic saline look fairly alike. hollow and black dots need further explanation.
We believe the reviewer was referring to the previous manuscript Figure 4 (now Figure 3), since hollow and black dots were mentioned. Although the MUs (dots) of each group present a similar distribution between hypertonic and isotonic conditions, there are differences that are supported by the statistical analysis. These differences are most noticeable between pre and post epoch, where the MUs re-arrangement occurs. Additionally, this is more easily observed at the Firing Rate plots.
What was the threshold to set it as black? How justified?
The motor units were assigned to each group (G1, hollow, G2, black) based on a K-means clusterization analysis, with three variables: RT, FR and MUAP amplitude (lines 130-132). This means that each group was defined by points with the least Euclidian distance between them, in a 3-dimensional configuration (due to the number of variables used).
One would like to see an association analysis of individual VAS versus firing rate, threshold and amplitude.
We thank the reviewer for the interesting suggestion. The VAS was used as a tool to measure and characterize the participant’s reported pain levels. Such an analysis, as suggested by the reviewer, would be an interesting topic of discussion on how a VAS correlates with the MU changes. However, we believe that would be out of the scope of this study, and would add an additional layer of analysis that could change the focus of our results on MU re-organization in response to induced muscle pain.
Were there differences between male and female subjects?
This question was also raised by the first reviewer. After submitting this manuscript, we became aware of the work of Lulic-Kuryllo & Inglis (published October 2022, doi: 10.1016/j.jelekin.2022.102689) which reported significant differences between male and female in terms of Firing Rate and MUAP amplitudes. As such, and to avoid data heterogeneity, we decided to remove the female group from this study. As this has reduced our sample size, we also include now a statistical power analysis to ensure the reliability of results.
Round 2
Reviewer 3 Report
The authors have addressed most of my comments. The presentation is improved.